# Coping Strategies in Elderly Colorectal Cancer Patients

**DOI:** 10.3390/cancers14030608

**Published:** 2022-01-26

**Authors:** Keyla Vargas-Román, María Isabel Tovar-Gálvez, Antonio Liñán-González, Guillermo Arturo Cañadas de la Fuente, Emilia Inmaculada de la Fuente-Solana, Lourdes Díaz-Rodríguez

**Affiliations:** 1Research Group CTS1068, Andalusia Research Plan, Junta de Andalucía, 18014 Granada, Spain; matoga@ugr.es (M.I.T.-G.); cldiaz@ugr.es (L.D.-R.); 2Spanish Education Ministry Program FPU16/01437, Methodology of Behavioral Sciences Department, Faculty of Psychology, University of Granada, 18071 Granada, Spain; 3Department of Nursing, Faculty of Health Sciences, Ceuta Campus, University of Granada, 51001 Ceuta, Spain; 4Department of Nursing, Faculty of Health Sciences, Melilla Campus, University of Granada, 52005 Melilla, Spain; antoniolg@ugr.es; 5Department of Nursing, Faculty of Health Sciences, University of Granada, 18016 Granada, Spain; gacf@ugr.es; 6Methodology of Behavioral Sciences Department, Faculty of Psychology, University of Granada, 18071 Granada, Spain; edfuente@ugr.es

**Keywords:** colorectal cancer, coping strategies, quality of life

## Abstract

**Simple Summary:**

Coping strategies help to mitigate the impact of a situation on an individual’s life. Elderly patients that undergo the stressful situation of going through colorectal cancer are no exception. This systematic review describes the improvement of the quality of life of patients who applied coping strategies to their situation. The results expose that coping strategies helped these patients to adapt and overcome the disease’s stressful scenarios.

**Abstract:**

In Spain, 34,331 new cases of colorectal cancer were diagnosed in 2018 and 15,923 individuals died from this disease in the same year. The highest incidence of colorectal cancer is among individuals aged 65–75 years and the physiological consequences of aging, alongside the effects of the disease and its treatment, can exacerbate their physical deterioration and cognitive impairment and reduce their social relationships. The learning of coping strategies may help to improve the quality of life of patients after cancer diagnosis. To test the hypothesis that the utilization of coping strategies can improve the quality of life of elderly patients with colorectal cancer, PubMed and EBSCO databases were searched, up to 2021, using the following terms: “coping strategies and colorectal cancer” with “anxiety”, “quality of life”, “depression”, “unmet needs”, “optimism”, “intimacy”, “distress”, “self-efficacy” and “self-esteem” with Boolean operators “AND”, “OR”. The literature search retrieved 641 titles/abstracts written in English. After an exhaustive analysis, only 7 studies met the inclusion criteria. Randomized evidence was scant and was reported only in 3/7 of the studies analyzed. Data from available randomized evidence support that patients improved on their depression and quality of life and felt more prepared to deal with their cancer. Coping strategies in patients with colorectal cancer were effective in improving patient adaptation to their new situation. Healthcare professionals working with these patients should receive training in this complementary treatment, to be able to conduct comprehensive care in order to improve the quality of life of these patients.

## 1. Introduction

According to the World Health Organization, cancer is the leading cause of death worldwide and there are around 18.1 million new cases every year, expected to increase to 24 million cases by 2035 [1,2]. The most frequently diagnosed cancers worldwide are lung (1.82 million), breast (1.67 million) and colorectal (1.36 million) cancers. The latter is the third most frequent cancer in males (10.0% of the total) and the second in females (9.2%), although there has been an increase in five-year survival rates due to diagnostic and therapeutic advances [3,4]. Therefore, cancer patients have increasingly become chronic patients, with the corresponding rise in health care costs [5].

In Spain, 34,331 new cases of colorectal cancer were diagnosed in 2018 and 15,923 individuals died from this disease in the same year [2,6]. Besides genetic factors, the development and prognosis of this disease is influenced by lifestyle factors, including obesity, physical inactivity and diet [7]. The highest incidence of colorectal cancer is among individuals aged 65–75 years and the physiological consequences of aging, alongside the effects of the disease and its treatment (chemotherapy, radiotherapy and surgery), can exacerbate their physical deterioration [8,9] and cognitive impairment and reduce their social relationships [10]. This situation increases the likelihood of anxiety and depression [11], further limiting activities of daily living and the quality of life [12].

The response to cancer diagnosis is generally characterized by three phases. In an initial denial phase, a resistance to believe the diagnosis is accompanied by emotional “anesthesia”. In a second phase, slow acknowledgement of the situation is accompanied by the emergence of anxiety, depression, insomnia and lack of appetite, among other symptoms.

The learning of coping strategies may help to improve the quality of life of patients after cancer diagnosis. Coping refers to the capacity of individuals to face stressing factors and manage their emotions [13] and there are two main types of coping strategy, one focused on the problem and the other on emotions [14]. In the former, patients center on facing their situation and changing their environment to ease their stress [15]. In contrast, the aim of the coping strategy centered on emotions is to seek the meaning of the stressing situation. In the “problem” approach, the person feels that something constructive can arise from the stress, whereas the emotional approach encourages the feeling that it can be endured [13].

Both types of coping strategy have been studied in women with chronic diseases such as fibromyalgia, including planning, distraction techniques and social support [16]. Other authors have studied a variety of approaches, including spirituality, family assistance/support, support groups and the promotion of a positive attitude [17]. In a study of patients with endometriosis, the authors described dietary changes, increased physical exercise, massage, meditation, acupuncture and Chinese medicine as the most frequently applied coping strategies, stressing that patients must first become aware of their disease [18]. Another study identified three types of self-care strategies employed by diabetic patients, namely, “proactive”, adopting a healthy lifestyle; “passive”, taking control of some aspects of self-care; and “non-conformists”, not managing to achieve a healthy lifestyle [19]. The most frequent coping strategies used by women with breast cancer receiving chemotherapy were reported to be self-affirmation and positive behaviors, with a preference for problem-centered rather than emotion-centered coping strategies [20].

With the above background, the objective of this study is to test the hypothesis that the utilization of coping strategies can improve the quality of life of elderly patients with colorectal cancer.

## 2. Materials and Methods

### 2.1. Search of the Literature and Study Selection

PubMed and EBSCO databases were searched, up to 2021, using the following terms: “coping strategies and colorectal cancer” with “anxiety”, “quality of life”, “depression”, “unmet needs”, “optimism”, “intimacy”, “distress”, “self-efficacy” and “self-esteem” with Boolean operators “AND”, “OR”.

### 2.2. Inclusion Criteria

The inclusion criteria were: publication in English; randomized control trial (RCT), controlled trial (CT), or single-group (quasi-experimental) design; study of patients aged ≥65 years diagnosed with any stage of colorectal cancer using coping strategies to improve associated symptoms/situations; and study of physical, psychological and/or behavioral measures (e.g., anxiety, depression, quality of life, unmet needs, privacy, anguish, optimism, self-efficacy, self-esteem, etc.).

### 2.3. Study Selection

This systematic review was registered in the International Prospective of Systematic Reviews (the study has been registered at PROSPERO; registration ID to be obtained) and conducted in accordance with the PRISMA Declaration [21]. The process of selection of the articles is illustrated in the PRISMA flowchart below. Starting with the screening stage, the titles and abstracts of retrieved articles were studied, followed by a reading of the whole text and the selection of studies meeting the inclusion criteria, eliminating duplicates (Figure 1).

### 2.4. Quality of Studies

The quality of RCTs was evaluated using the validated scale of Jadad [22], which has three components, i.e., (1) individuals randomly selected for the experimental group (score of 0–2); (2) double-blinded intervention (score of 0–2); (3) adequate description of subjects not completing the study (score of 0–1). Studies with scores ≥3 were considered to be of high quality. As reported in Table 1, all three RCTs selected [23,24,25] specified the randomization method and included a flow diagram explaining when and why patients were lost to the study, obtaining 3 out of 5 points, i.e., maximum design quality.

## 3. Results

### 3.1. Description of the Studies

The literature search retrieved 641 titles/abstracts written in English. Eligibility criteria were met by seven studies, three RCTs [23,24,26], two quasi-experimental studies [27,28] and two controlled clinical trials [25,29]. The main reasons for exclusion were: study variables not in agreement with review objectives; descriptive study alone, with no evaluation of the effect of interventions; or non-reporting of results (presentation of controlled clinical trial protocols). Table 1 summarizes the results obtained in the reviewed articles.

### 3.2. Participants

Four of the seven selected studies were carried out in Australia, two in the USA (North Carolina and California) and one in Canada. The mean age of the participants involved in the studies was 65 years old. The patients were undergoing oncological treatment [23,24,26,29] or were post-treatment survivors [25,27,28].

Regarding the number of participants, the studies varied in numbers. One studied 74 participants [26], another studied 111 [29]; one studied 356 [24] and another 1164 [23], while the remaining three studies had samples of <50 individuals [25,27,28]. In relation to gender, females formed the majority of the sample in three of the studies [23,24,26]; there was a similar proportion of males and females in another three studies [27,28,29] and the remaining study focused on only males [25].

### 3.3. Types of Intervention

Five studies used individual phone calls to assess the most important needs of patients [23,24,25,27,29] and the remaining two used coping strategies with patients in person, addressing sexual and cognitive aspects [26,28].

Telephone interventions included the “Pathfinder Program”, using volunteers trained for three days on knowledge of colorectal cancer and the needs of the patients and on related social support and healthcare services [29]; the “CONNECT” program, adapted for patients with colorectal cancer and designed to identify their main needs after hospital discharge and enable the provision of quality care [25]; an adapted information-based program [23]; two programs that aimed at reducing anxiety and depression levels were compared, after assessing the needs of patients [24]; and a study that included three follow-up calls to evaluate the response of survivors to an information package (DVD, leaflets and questionnaires), with a care plan and nursing treatments [27].

With respect to in-person interactions, a study reported on a “meaning-making” program in which patients narrated their life before, during and after cancer [26]. Furthermore, in another study, patients and their partners were invited to evaluate their sex-related needs after colorectal cancer diagnosis [28]. These two interventions were conducted by oncological nurses and healthcare professionals.

### 3.4. Effect of Interventions

In general, work on coping strategies resulted in significant positive changes in patients with colorectal cancer patients, with improvements in anxiety and depression [24,27,29], quality of life [24,25,27], emotional distress [25,27,28] and self-efficacy [23,26].

### 3.5. Professionals Conducting Interventions

Most interventions were largely conducted by nurses specialized in the care of patients with colorectal cancer [23,24,25,27], although patients were sometimes referred to other specialists when deemed necessary [25]. The “meaning-making” coping program was conducted by psychologists to improve different cognitive and emotional domains [26]. In the remaining two studies, the program was run by professionals who were specifically trained according to the needs detected in patients [28,29].

### 3.6. Intervention Duration, Frequency and Intensity

The coping programs had a mean duration of 6 months [24,25], ranging from one week [28], to two-three months [27,29] and one year [23,26]. Few studies specified the frequency of sessions, which was twice weekly [27], a single session alone [23] and a maximum of four sessions [26]. The duration of each session also varied widely from 15–20 min [25], to 50 min [28], 60–90 min [27] and 2 h [26].

### 3.7. Variables

The studies measured different variables related to psychological status, such as sexual concerns [28]; unmet needs [27]; anxiety and depression [24,29]; self-esteem, optimism and self-efficacy [26]; knowledge of preparation levels and health recommendations [23]; or quality of life [25,27]. As measurement instruments, three studies used the “Hospital Anxiety and Depression Scale (HADS) [24,27,29] and Supportive Care Need Scale (SCNS) or its equivalent (SNSN) [24,25,29], while a study did not specify the measurement instrument used [23].

All studies analyzed short-term endpoints except for two that also considered longer-term outcomes at 1, 3 and 7 weeks post-intervention. Another study considered a follow-up every 6 weeks post-intervention [24,27]. One study did not specify the follow-up period studied [29] (see Table 1).

## 4. Discussion

This review found that the use of coping strategies as a complementary approach to conventional cancer therapy in patients with colorectal cancer significantly improved their symptoms of depression, anxiety, self-efficacy and distress and enhanced their quality of life. Three out of the seven studies in the review were RCTs, offering the highest quality of evidence. Randomized studies showed that participants in the experimental intervention group made improvements and strengthened areas in their lives that helped them to better cope with their cancer. Including getting better with depression, feeling more informed and prepared and improving their self-esteem, this added up to a better quality of life for patients suffering from colorectal cancer in later life.

Consequently, improvements were observed in all seven studies, which varied in the type of coping strategy program, in its frequency and duration and in the variables examined. Interventions were adapted to the particular needs identified for each patient, preventing recommendation of a standard coping strategy that was the most effective to treat colorectal cancer patients.

Different study variables were considered using various measurement instruments. Two studies used the same instrument (HADS questionnaire) to measure anxiety and depression [24,29]. The quality of life was evaluated in two studies [24,25] with questionnaires of the European Organization for Research and Treatment of Cancer (QLQ and QLQ-C30) and, in another, with the Functional Assessment of Cancer Therapy (FACT-C) [27]. Support care needs were assessed in two studies with the Supportive Care Needs Survey (SCNS) [25,29] and, in another study, with the Supportive Needs Survey (SNS) [24]. The timing of evaluations also varied widely, being carried out at three months in three studies [27,28,29], at six months in two [24,25] and at one year in the remaining two studies [23,26].

All interventions were aimed at individual patients [24,25,26,27,29] with the exception of one of the programs in which included couples who had lived together for at least one year [28] and a study that differentiated interventions by ethnic group [23].

Some limitations found were, such as that only a small number of studies met the review inclusion criteria and that they widely differed in the frequency and duration of programs, as well as in the endpoints analyzed. Due to the scarcity of randomized studies in this setting, this highlights the difficulties in giving firm results. Another matter to be discussed is the possibility of biases from non-randomized, quasi-experimental and cohort studies, since, compared with randomized studies, the latter showed evidence of superior quality. Nonetheless, the sample-size was a major issue in ¾ of randomized studies. Thereafter, no firm conclusion may be given in any setting until adequate powered randomized trials are available. In addition, interventions were adapted to the needs of individual patients, hampering the drawing of general conclusions. The results of this review point to the need for further RCTs on coping strategies in more homogeneous samples of patients with colorectal cancer, using validated questionnaires or scales, in order to establish the optimal approach.

## 5. Conclusions

The learning of coping strategies can reduce distress, depression and anxiety in patients diagnosed with colorectal cancer, improving their quality of life. Healthcare professionals working with these patients, especially nurses, should receive training in this complementary treatment. Further research is warranted to identify coping mechanisms that are the most effective in this patient population.

## Figures and Tables

**Figure 1 cancers-14-00608-f001:**
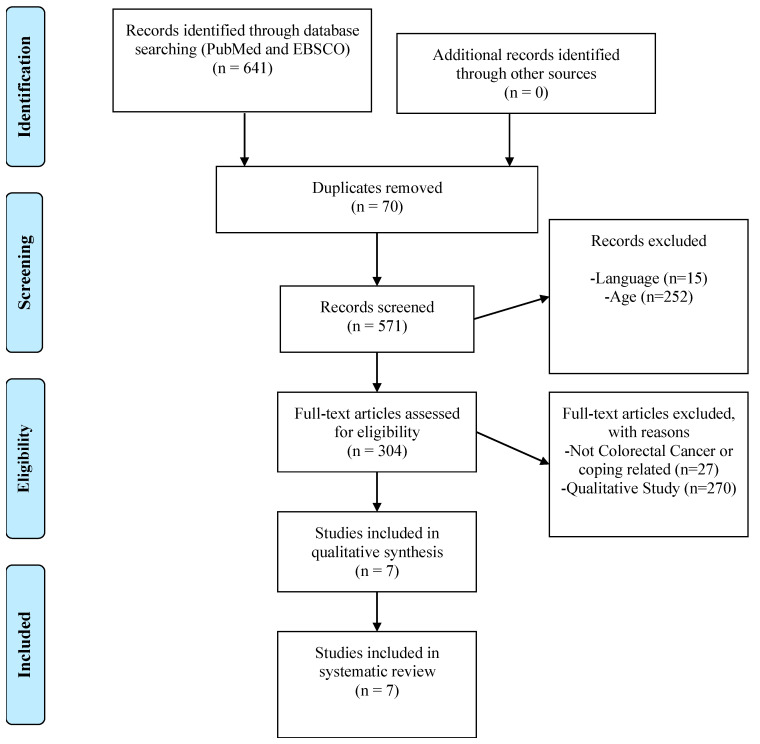
Flowchart of the selected studies.

**Table 1 cancers-14-00608-t001:** Characteristics of the included studies.

Author	Study Design	Participants	Type and Duration of the Intervention	Measurement Instruments	Results
**Jerant (2014)**	Randomized Controlled Trial	A total of 1164 colorectal cancer patients grouped by ethnic group/language (49.3% non-Hispanic, 27.2% Hispanic/English, 23.4% Hispanic/Spanish). Experimental group (EG): 595 participants receiving adapted program (IMPC). Control group (CG): 569 participants receiving non-adapted program.	Duration of 12 months. The EG received a program adapted to their corresponding ethnic group to measure their knowledge of colorectal cancer, among other variables. The CG received the non-adapted program. Levels of knowledge were evaluated in visits before and after the intervention.	Interactive multimedia computer program (IMCP) that screened knowledge, self-efficacy, test preference specificity, discussion and recommendation.	Significant improvement versus the control group in knowledge (*p* < 0.001), self-efficacy (*p* < 0.01), preparation (*p* < 0.05), test preference specificity (*p* < 0.01), discussion (*p* < 0.01) and recommendation (*p* < 0.05); 95% CI, *p* < 0.05.
**Barsky (2012)**	Quasi-Experimental Study	Nine individuals with sexual problems after colorectal cancer and their partners in a total of eighteen subjects (cohabiting for >1 year).	Duration of 50 min/day for one week. Sexual concerns of colorectal cancer patients and their partners were evaluated over telephone.	Sexual Distress (ISS), higher ISS scores indicate greater degree of distress; Female Sexual Function (FSFI); Sexual Communication (DSCS); Dyadic Adjustment (DAS-4) and Intimacy (MSIS).	Patient data showed large effect size (≥0.80) for sexual distress, female sexual function and sexual communication; medium effect size (0.30–0.60) for dyadic adjustment; and small effect size (0.20–0.30) for intimacy (Cohen’s size effects range).
**Jefford (2011)**	Quasi-Experimental Study	Ten survivors of colorectal cancer.	Total duration of 7 weeks. Participants received an information pack including DVD and leaflets. Nursing interventions (duration of 60–90 min).	Brief Symptom Inventory (BSI-18), Cancer Survivors Unmet Needs (CaSUN) and European Organization for Research and Treatment of Cancer (QLQ-C30).	In total, 70% of the subjects showed no distress during post-intervention follow-up. Unmet needs decreased from an average of 7 to an average of 4 during the follow-up. The quality of life remained high at both baseline and during follow-up.
**Young (2009)**	Controlled Clinical Trial	A total of 41 survivors of colorectal cancer admitted to the Royal Prince Alfred Hospital (Sydney, Australia), with 21 in the control group and 20 in the experimental groups.	Duration of 6 months post-discharge. CONNECT telephone-based intervention. Duration of each call was 14–19 min.	Distress Thermometer and Supportive Care Needs Survey (SCNS)	On the third month post-intervention, the SCNS showed improvement in psychological support (*p* = 0.05), support from health system and information (*p* = 0.001), physical and daily activities (*p* = 0.04) and patient care and support (*p* = 0.002).
**Girgis (2009)**	Randomized Controlled Trial	A total of 356 patients under treatment for colorectal cancer, randomly divided into three groups, Usual Care (CG, n = 117), Telephone Caseworker (TCW, n = 20) and Oncologist/General practitioner (O/GP, n = 119).	Duration of 6 months. Evaluation at 0, 3 and 6 months.	Hospital Depression and Anxiety Scale (HADS), European Organization for Research and Treatment of Cancer (QLQ-C30), Supportive Needs Survey, Needs Assessment for Advanced Cancer Patient Questionnaire (NA-ACP).	The TCW group evidenced reduced depression (HADS) (*p* = 0.01), improved physical functions (QLQ-30) (*p* = 0.01), fewer unmet needs (*p* = 0.07), better discussion of their situation (*p* < 0.0001) and improvement in communication with their medical care team (*p* = 0.0005) (NA-ACP); 95% CI, *p* < 0.05.
**Macvean et al. (2006)**	Controlled Clinical Trial	n = 52 patients under treatment for colorectal cancer. Experimental group (EG) = 18; control group (CG) = 34.	Pathfinder Intervention Program (phone-based) EG: Adaptation of strategies to needs detected in each patient. CG: standard care. Evaluations before and 3 months after intervention.	Hospital Anxiety and Depression Scale (HADS) and 59-item Supportive Care Needs Survey (SCNS).	The EG showed a significant improvement versus the CG in depression (HADS) (*p* = 0.011) and care support needs (SCNS) (*p* = 0.004).
**Lee (2006)**	Randomized Controlled Trial	N = 74 patients under treatment for colorectal cancer; experimental group (EG), 35 participants received “meaning-making” intervention (n = 35); control group (CG), participants received “usual” care intervention (n = 39).	Duration of 12 months with evaluations at 3, 6, 9 and 12 months. EG participants underwent up to four individualized interventions (120 min/session) at home or in the clinic (as preferred by patient), involving an exercise to guide participants through a narration of their experience with cancer. CG participants received usual care (information on the availability of community- and hospital-based programs and psychological support).	Rosenberg Self-Esteem Scale (RSES), Life Orientation Test-Revised (LOT-R), Generalized Self-Efficacy Scale (GSES).	The EG showed improvements versus the control group in self-esteem (RSES) (*p* = 0.006), optimism (LOT-R) (*p* = 0.019) and self-efficacy (GSES) (*p* = 0.002); 95% CI, *p* < 0.05.

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
