# Peer review of "Coping Strategies in Elderly Colorectal Cancer Patients"

_cancers, 2022, doi:10.3390/cancers14030608_

Round 1

Reviewer 1 Report

Manuscript is interesting and underscoring the paucity of data availble in this clinical envirorment. So the quality of data and related biases should be better depicted

Abstract

Randomized evidence was scant and was reported only in 3/7 of the study analyzed.

Data from available randomized evidence support that ... (p.).

Results

Data from randomized evidence should be summarized in a table indicating each analyzed outcome and relative results and 95%CI and p.

Discussion should clearly depict the scarcity of studies in this setting and the difficulties in giving firm results

Data from randomized trial and trial may be discuss in 1-3 more phrases

Possibility of biases from non randomized quasi experimental and cohorts studies should be underscored

Overall from many outcomes there is evidence that coping strategies might be of value in some settings.

Nonetheless the sample-size is a major issue in ¾ of randomized studies. Thereafter no firm conclusion may be given in any setting until adequate powered randomized trials will be available.

Author Response

Dear Reviewers,

Thank you for the comments regarding the manuscript, we took the comments into consideration and modified the manuscript accordingly. Hope this answers all the concerns presented. If any other concern is detected please don’t hesitate to ask.

Best regards,

Keyla Vargas

Reviewer #1

Manuscript is interesting and underscoring the paucity of data availble in this clinical envirorment. So the quality of data and related biases should be better depicted

Abstract (Added the sentences accordingly)

Randomized evidence was scant and was reported only in 3/7 of the study analyzed. (added)

Data from available randomized evidence support that ... (p.). (added)

Results

Data from randomized evidence should be summarized in a table indicating each analyzed outcome and relative results and 95%CI and p. (added the CI the p values were on the table)

Discussion should clearly depict the scarcity of studies in this setting and the difficulties in giving firm results (added for clarification)

Data from randomized trial and trial may be discuss in 1-3 more phrases (phrases added)

Possibility of biases from non randomized quasi experimental and cohorts studies should be underscored (noted and added to the discussion)

Overall from many outcomes there is evidence that coping strategies might be of value in some settings.

Nonetheless the sample-size is a major issue in ¾ of randomized studies. Thereafter no firm conclusion may be given in any setting until adequate powered randomized trials will be available. (added to the limitations)

Reviewer 2 Report

The Authors have conducted an systematic review to address a very important question, i.e. how effective are the coping strategies applied by elderly patients with CRC. The topic is of utmost importance.

Unfortunately, I have major difficulties in understanding how the studies were selected for inclusion. The literature research dates back to Sept 2019 (in the abstract; this is Aug 2019 in the MS body, and this small inconsistence should be reconciled). I have carried an explorative search using the terms reported in the methods, and come to completely different results. Furthermore, the flow chart does not report how many studies were excluded since they were not focused on elderly patients - and I cannot find the mean age of the patients involved in the selected studies anywhere in the paper.

Can the Authors fully reconsider the study selection, or make it clearer, and rework the paper, including the Discussion, accordingly?

Minor comments:

  • Lines 17-20 in the abstract can be removed.
  • the same can be applied to lines 37-48.
  • Institutional Review Board Statement and Acknowledgments were not filled

Author Response

Dear Reviewers,

Thank you for the comments regarding the manuscript, we took the comments into consideration and modified the manuscript accordingly. Hope this answers all the concerns presented. If any other concern is detected please don’t hesitate to ask.

The Authors have conducted a systematic review to address a very important question, i.e. how effective are the coping strategies applied by elderly patients with CRC. The topic is of utmost importance.

Unfortunately, I have major difficulties in understanding how the studies were selected for inclusion. The literature research dates back to Sept 2019 (in the abstract; this is Aug 2019 in the MS body, and this small inconsistence should be reconciled). I have carried an explorative search using the terms reported in the methods, and come to completely different results. (Updated the search to the latest date in which was reviewed)

Furthermore, the flow chart does not report how many studies were excluded since they were not focused on elderly patients. (The flowchart was redone with the latest results and added the excluded studies from the age in the patients involved.)  - and I cannot find the mean age of the patients involved in the selected studies anywhere in the paper (the mean age was added in the results part of the manuscript.)

Can the Authors fully reconsider the study selection, or make it clearer, and rework the paper, including the Discussion, accordingly?

Minor comments:

  • Lines 17-20 in the abstract can be removed.
  • the same can be applied to lines 37-48.
  • Institutional Review Board Statement and Acknowledgments were not filled

Round 2

Reviewer 2 Report

The Authors have reviewed most of my comments, please let me thank them for that.

(very) minor: Institutional Review Board Statement and Acknowledgments are not filled in.